# Correlation between hospitalized patients' demographics, symptoms, comorbidities, and COVID-19 pandemic in Bahia, Brazil

**Márcio C. F. Macedo**[1]*, **Isabelle M. Pinheiro**[2,3], **Caio J. L. Carvalho**[1,4], **Hilda C. J. R. Fraga**[3,5], **Isaac P. C. Araujo**[6], **Simone S. Montes**[3], **Otávio A. C. Araujo**[1], **Lucas A. Alves**[6], **Hugo Saba**[3,7], **Márcio L. V. Araújo**[8,3,9], **Ivonete T. L. Queiroz**[10], **Romilson L. Sampaio**[3,11], **Márcia S. P. L. Souza**[9], **Ana Claudia F. N. da Silva**[9], **Antonio C. S. Souza**[3,11]

1 Department of Computer Science, Federal University of Bahia, Salvador, Bahia, Brazil, 2 Department of Radiology, Federal Institute of Bahia, Salvador, Bahia, Brazil, 3 Graduate Program in Knowledge Diffusion, Federal University of Bahia, Salvador, Bahia, Brazil, 4 Supercomputing Center for Industrial Innovation, SENAI CIMATEC, Salvador, Bahia, Brazil, 5 Oswaldo Cruz Foundation, Salvador, Bahia, Brazil, 6 Department of Electrical Engineering, Federal University of Bahia, Salvador, Bahia, Brazil, 7 Department of Exact and Earth Sciences, University of the State of Bahia, Salvador, Bahia, Brazil, 8 Department of Technical Education, Federal Institute of Bahia, Santo Amaro, Bahia, Brazil, 9 Health Secretary of the State of Bahia, Salvador, Bahia, Brazil, 10 Department of Technical Education, Federal Institute of Bahia, Barreiras, Bahia, Brazil, 11 Department of Computer Science, Federal Institute of Bahia, Salvador, Bahia, Brazil

* marcio.cerqueira@ufba.br

## Abstract

In this paper, we provide a retrospective cohort study with patients that have been hospitalized for general or intensive care unit admission due to COVID-19, between March 3 and July 29, 2020, in the state of Bahia, Brazil. We aim to correlate those patients' demographics, symptoms and comorbidities, with the risk of mortality from COVID-19, length of hospital stay, and time from diagnosis to definitive outcome. On the basis of a dataset provided by the Health Secretary of the State of Bahia, we selected 3,896 hospitalized patients from a total of 154,868 COVID-19 patients that included non-hospitalized patients and patients with invalid registration in the dataset. Then, we statistically analyzed whether there was a significant correlation between the patient record data and the COVID-19 pandemic, and our main findings reinforced by the use of a multivariable logistic regression were that older age (Odds Ratio [OR] = 1.03, 95% Confidence Interval [CI] = 1.03-1.04, p-value ($p$) <0.001), an initial symptom of shortness of breath (OR = 1.88, 95% CI = 1.60-2.20, $p$ < 0.001), and the presence of comorbidities, mainly chronic kidney disease (OR = 2.41, 95% CI = 1.67-3.48, $p$ < 0.001) are related to an increased risk of mortality from COVID-19. On the other hand, sore throat (OR = 0.74, 95% CI = 0.58-0.95, $p$ = 0.02) and length of hospital stay (OR = 0.96, 95% CI = 0.58-0.95, $p$ < 0.001) are more related to a reduced risk of mortality from COVID-19. Moreover, a multivariable linear regression conducted with statistically significant variables ($p$ < 0.05) showed that age (OR = 0.97, 95% CI = 0.95-0.98, $p$ < 0.001) and time from diagnosis to definitive outcome (OR = 1.67, 95% CI = 1.64-1.71, $p$ < 0.001) are associated with the length of hospital stay.

**Data Availability Statement:** All relevant data are within the manuscript and its Supporting information files.

**Funding:** Hugo Saba received financial support from the National Council for Scientific and Technological Development - CNPq (http://cnpq.br/), grant numbers 431990 / 2018-2 and 313423 / 2019-9 Márcio C. F. Macedo received financial support from the Postdoctoral National Program of the Coordination for the Improvement of Higher Education Personnel - CAPES (https://www.capes.gov.br/), grant number 88882.306277/2018-01. This work was supported by the National Council of Institutions of the Federal Network of Vocational, Scientific and Technological Education - CONIF (http://portal.conif.org.br/br/). This work was also supported by the Secretariat of Professional and Technological Education - SETEC (http://portal.mec.gov.br/setec-secretaria-de-educacao-profissional-e-tecnologica). The funders had no role in study design, data collection and analysis, decision to publish, or preparation of the manuscript.

**Competing interests:** The authors have declared that no competing interests exist.

## Introduction

The beginning of 2020 was marked by the emergence of a mysterious pneumonia caused by a variation of the coronavirus, after the first case was reported in December 2019 in Wuhan, China. With the rapid increase in cases, the World Health Organization (WHO) declared in 2020 the severe acute respiratory syndrome coronavirus 2 (SARS-CoV-2) as a public health emergency of international interest [1].

The SARS-CoV-2 that causes the coronavirus disease 2019 (COVID-19) arises from the RNA mutation of a virus isolated for the first time in 1937 and which initially expanded asymptomatically and with mild symptoms. However, SARS-CoV-2 is currently the pathogen that causes the most concern to the community around the world due to its high transmissibility, thus generating major impacts for health systems [1].

Faced with a still obscure and uncertain scenario, WHO declared COVID-19 as a pandemic on March 11, 2020 and instituted essential measures to prevent and cope with the disease, which included hand hygiene with soap and water, or gel alcohol in situations where the use of soap and water was not possible. WHO also recommended the entire society to avoid touching eyes, nose and mouth, to protect the people around, when sneezing or coughing, in addition to the use of the mask and the implementation of the distance measure of at least one meter, thus avoiding crowds [1].

In Brazil, after the pandemic spread to several countries, the COVID-19 infection was declared by the Ministry of Health, on February 3, 2020, a public health emergency of national importance. This action aimed to encourage the taking of administrative measures with greater agility so that the country could begin to prepare itself to face the pandemic, whose first case of COVID-19 was notified only on the day February 26, 2020, in São Paulo [2].

Within the perspective of adopting measures that would generate less impact on the Brazilian health system, the Ministry of Health authorized and defined, through ordinance, the setting up of Field Hospitals and the implementation of specific Health Units to meet the exclusive demands of patients with COVID-19, operating in the following regulated and structured ways: as a clinical inpatient unit for patients with low-complexity respiratory symptoms and as a pulmonary ventilatory support unit, for treating cases of worsening of the patient's respiratory condition, indicating the invasive and non-invasive ventilatory support [3].

The Brazilian National Health System (Sistema Único de Saúde—SUS) is one of the largest and most complex public health systems in the world and provides services ranging from a simple blood pressure assessment to a complex organ transplant. Its operation guarantees full, universal and free access for the entire Brazilian population, without discrimination.

However, despite all the recommendations provided by WHO and the Ministry of Health in Brazil, by the day of September 8, 2020, COVID-19 has already infected about 27.4 million people around the globe, leading to almost 900,000 deaths. In Brazil, the second most infected country to the date, 4.1 million residents were diagnosed with COVID-19, and nearly 127,000 deaths have been estimated [4]. The most common reported symptoms associated with COVID-19 were flu-like symptoms (*e.g.*, fever) and respiratory dysfunctions (*e.g.*, dyspnea), while the highest number of deaths happened in elderly patients or patients with a previous comorbidity [5].

The current increase in demands for hospital admissions creates an overload in both private and public health systems, which already operate at their capacity limit, that requires a careful structuring of SUS, to promote care to the population with agility and quality, given the magnitude of the COVID-19 pandemic. As a result, the assessment and correlation of the most common variables that could influence in the risk of mortality, length of hospital stay and time from diagnosis to definitive outcome of COVID-19 have become essential to aid health care

professionals in the decision-making process and to ensure that the health care services could be provided with the greatest possible safety, reliability and agility [6].

Several studies have already correlated patients' medical data and the COVID-19 pandemic, in order to detect the variables that could be useful to predict mortality risk [7–16], or to predict length of hospital stay [17, 18]. As stated by the systematic review of Wynants *et al.* [5], age, sex, the presence of comorbidities, and a few biomarkers, such as C reactive protein, creatinine, lymphocyte count, red cell distribution width, and lactate dehydrogenase, are commonly reported as variables correlated with an increased mortality risk, while age and variables derived from computed tomography scans are reported to be more correlated with the length of hospital stay. Using data from Brazil, the work of Soares *et al.* [19] reported that age and shortness of breath are the variables most related to the risk of mortality from COVID-19.

In this paper, we aim to complement the studies regarding the COVID-19 pandemic and provide a data exploratory analysis of the variables that could be influencing the COVID-19 pandemic in the state of Bahia, Brazil. In this sense, we aim to verify not only whether and how much patients' demographics, symptoms, and comorbidities are correlated with the risk of mortality from COVID-19, but also whether those variables are correlated with the length of hospital stay and the time from diagnosis to definitive outcome, correlation that is not much analyzed by related work. This study is useful not only to check whether the findings of related work are also valid for data collected in Brazil, but also to guide the health care professionals and the scientific community with respect to the characteristics of the COVID-19 pandemic in a state of Brazil.

## Materials and methods

### Data collection

In this retrospective study, we included patients that live in the state of Bahia, Brazil, and who were diagnosed with COVID-19, following the guidance provided by WHO, between March 3, 2020 and July 29, 2020. We used the dataset provided by the Health Secretary of the State of Bahia (Secretária de Saúde do Estado da Bahia—SESAB), which included anonymized data of patients' demographics (*e.g.*, birth date, sex, race, state of residence, municipality of residence), symptoms (*e.g.*, sore throat, shortness of breath, fever, cough), comorbidities (*e.g.*, diabetes, cardiovascular disease, chronic kidney disease, immunosuppression), patient record data collection (*e.g.*, diagnosis date, symptoms start date, hospital ID, hospitalization date, length of hospital stay, severity status, pregnancy), and outcome data (*e.g.*, whether the patient is dead, has recovered or is still under treatment, and the date of the definitive outcome). All of these data were reported by the own patients, by their families, or by the health care professionals working in the hospitals. In S1 Table, we provide the anonymized, processed dataset with hospitalized COVID-19 patients that we have used as a basis to report and discuss the results presented in this manuscript.

### Statistical analysis

In terms of statistical analysis, discrete variables (*e.g.*, age, length of hospital stay, time from diagnosis to definitive outcome) were reported using mean and standard deviations metrics. Categorical variables with binary values were reported using count (n) and also the relative percentage (%). One-way Analysis Of Variance (ANOVA), Pearson correlation, and Chi-Square test were used to estimate p-values (*p*) between a categorical and a discrete variable, between discrete variables, and between categorical variables, respectively. We have used a default threshold of $p < 0.05$ to indicate that a reported result is statistically significant, and included the statistically significant variables in a multivariable regression analysis, to account

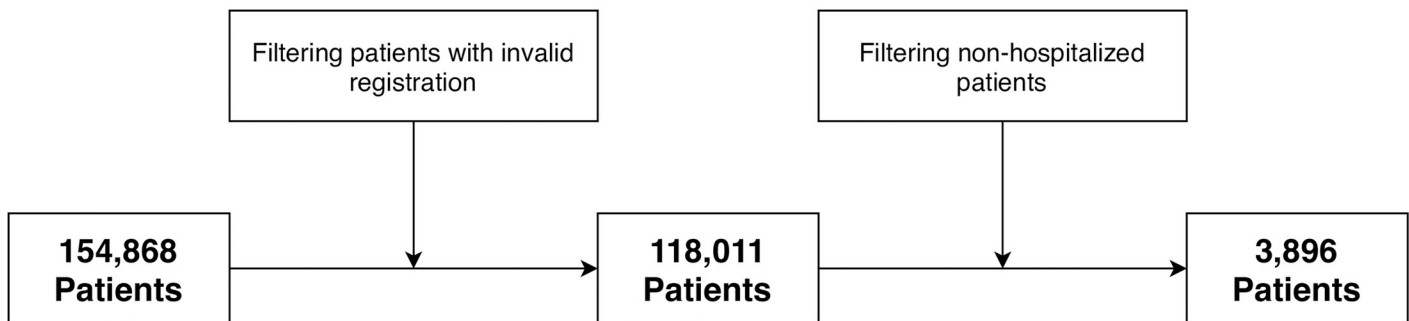

**Fig 1. Data processing overview.** The dataset provided by SESAB contained a total of 154,868 patients that were detected with COVID-19 in the state of Bahia, Brazil. After removing the patients with invalid data (*e.g.*, notification year prior to 2020, length of hospital stay lower than zero, age lower than zero, unknown outcome), the dataset contained a total of 118,011 patients with valid registration. Finally, we removed the patients that were not admitted to a hospital, restricting our evaluation to a final count of 3,896 patients.

for the influence that those variables might have on each other. In this case, odds ratio (OR), 95% confidence interval (CI), and new p-values are reported. Also, we have used popular libraries in the Python programming language (*e.g.*, pandas, numpy, statsmodels) to measure the reported metrics.

## Results

As can be seen in Fig 1, the table provided by SESAB contained data of 154,868 patients that were diagnosed with COVID-19 in the state of Bahia, Brazil. However, after doing some data processing and filtering to remove patients with invalid data or that were not hospitalized, 3,896 patients remained in the dataset. An overview of the valid data collected in this work, described according to the definitive outcome of the COVID-19 patients, is shown in Table 1.

According to Table 1, 26.8% of the hospitalized patients diagnosed with COVID-19 died from such a disease. In this case, the mean age of the hospitalized patients that died due to COVID-19 is higher than the mean age of the hospitalized patients that recovered from COVID-19 (68.8 years vs 57.7 years, $p < 0.001$). In terms of sex, although more male patients were diagnosed with COVID-19 (2,338 males vs 1,558 females), this factor was not determined to be statistically significant to be correlated with the risk of mortality from COVID-19 ($p = 0.8$). Likewise, race was found to be not statistically significant to be correlated with the risk of mortality from COVID-19 ($p = 0.1$). With respect to the COVID-19 symptoms, cough and sore throat were the most and the less common symptoms reported by the patients. On the other hand, shortness of breath and sore throat were the only ones whose p-values were below the pre-defined threshold. In this sense, shortness of breath was more common in patients that did not survive from COVID-19 (60.7% vs 40.6%, $p < 0.001$). On the other hand, sore throat was the less common reported symptom for both patients that died or survived from COVID-19 (10.4% vs 15.2%, $p < 0.001$). Given the comorbidities found in the dataset provided by SESAB, only immunosuppression ($p = 0.4$) and chromosomal disorder ($p = 0.7$) were found to be not statistically significant for our analysis. All the remaining comorbidities, namely diabetes (26.5% vs 19.0%, $p < 0.001$), chronic kidney disease (6.6% vs 3.0%, $p < 0.001$), chronic respiratory disease (6.8% vs 4.2%, $p = 0.002$), and cardiovascular disease (27.3% vs 21.1%, $p < 0.001$) were more frequent in non-survivor patients. Among female patients, low-risk pregnancy was more frequent in recovered patients (1.7% vs 0.4%, $p = 0.002$), meanwhile high-risk pregnancy was not significant enough to differentiate survivor

**Table 1. An overview of the dataset used in this study, according to the definitive outcome of hospitalized COVID-19 patients.**

| | Patients (n = 3,896) | | |
|---|---|---|---|
| **Variables** | **Died (n = 1,045)** | **Rec. (n = 2,851)** | **p-value** |
| Age | 68.8 (15.9) | 57.7 (18.9) | **< 0.001** |
| **Sex** | | | 0.8 |
| Female | 414 (39.6%) | 1,144 (40.1%) | |
| Male | 631 (60.4%) | 1,707 (59.9%) | |
| **Race** | | | 0.1 |
| Black | 682 (65.3%) | 1,803 (63.2%) | |
| White | 107 (10.2%) | 381 (13.4%) | |
| Asian | 147 (14.1%) | 370 (13.0%) | |
| Native American | 2 (0.2%) | 5 (0.2%) | |
| Unknown | 107 (10.2%) | 292 (10.2%) | |
| **Symptoms** | | | |
| Sore throat | 109 (10.4%) | 432 (15.2%) | **< 0.001** |
| Shortness of breath | 634 (60.7%) | 1,157 (40.6%) | **< 0.001** |
| Fever | 590 (56.5%) | 1,655 (58.0%) | 0.4 |
| Cough | 700 (67.0%) | 1,921 (67.4%) | 0.8 |
| **Comorbidities** | | | |
| Diabetes | 277 (26.5%) | 542 (19.0%) | **< 0.001** |
| Immunosuppression | 34 (3.3%) | 76 (2.7%) | 0.4 |
| Chronic kidney disease | 69 (6.6%) | 86 (3.0%) | **< 0.001** |
| Chronic respiratory disease | 71 (6.8%) | 121 (4.2%) | **0.002** |
| Cardiovascular disease | 285 (27.3%) | 601 (21.1%) | **< 0.001** |
| Chromosomal disorder | 12 (1.1%) | 27 (0.9%) | 0.7 |
| **Others** | | | |
| Low-risk pregnancy | 4 (0.4%) | 49 (1.7%) | **0.002** |
| High-risk pregnancy | 3 (0.3%) | 11 (0.4%) | 0.9 |
| Length of hospital stay | 10.1 (9.1) | 16.1 (11.7) | **< 0.001** |
| Time from diagn. to outc. | 17.2 (12.7) | 24.7 (12.6) | **< 0.001** |

Discrete variables are reported using the pattern: mean (standard deviation). Categorical binary variables are reported using the pattern: count (relative percentage with respect to the total number of deaths or recoveries). Age is reported in years. Length of hospital stay and time from diagnosis to definitive outcome are reported in days. One-way ANOVA was used to estimate p-values for age, length of hospital stay and time from diagnosis to definitive outcome variables. The remaining p-values were estimated using the Chi-Square test. Immunosuppression may refer to a patient with HIV infection or autoimmune disease. Bold p-values indicate statistically significant p-values lower than 0.05. "Rec." is an abbreviation for "Recovered". "Time from diagn. to outc." is an abbreviation for "Time from diagnosis to definitive outcome".

and non-survivor patients ($p = 0.9$). In this case, we note that 91.4% of the low-risk pregnants diagnosed with COVID-19 and available in our dataset have been hospitalized. Since we do not know in which month of their pregnancy these patients have been hospitalized, or whether the hospitalizations have been performed as a prevention measure to protect the patients, we believe that this high percentage of hospitalization may explain their high recovery rate. Finally, the recovered patients demanded more days of hospitalization (16.1 days vs 10.1 days, $p < 0.001$) and time from diagnosis to definitive outcome (24.7 days vs 17.2 days, $p < 0.001$) than the patients who died from COVID-19.

The statistically significant variables estimated from Table 1 were included into a multivariable logistic regression analysis, whose results are reported in Table 2. As we can see from that

**Table 2. Risk factors associated with the mortality of hospitalized COVID-19 patients.**

| Variables | OR (95% CI) | p-value |
|---|---|---|
| Age | 1.03 (1.03–1.04) | **< 0.001** |
| **Symptoms** | | |
| Sore throat | 0.74 (0.58–0.95) | **0.02** |
| Shortness of breath | 1.88 (1.60–2.20) | **< 0.001** |
| **Comorbidities** | | |
| Diabetes | 1.16 (0.96–1.40) | 0.1 |
| Chronic kidney disease | 2.41 (1.67–3.48) | **< 0.001** |
| Chronic respiratory disease | 1.12 (0.79–1.58) | 0.5 |
| Cardiovascular disease | 0.94 (0.78–1.14) | 0.5 |
| **Others** | | |
| Low-risk pregnancy | 0.48 (0.15–1.56) | 0.2 |
| Length of hospital stay | 0.96 (0.95–0.97) | **< 0.001** |
| Time from diagnosis to definitive outcome | 0.97 (0.96–0.98) | **< 0.001** |

Multivariable logistic regression was used to measure odds ratios (OR), confidence intervals (CI) and p-values for the statistically significant variables measured in Table 1 with respect to the definitive outcome of hospitalized COVID-19 patients. Bold p-values lower than 0.05 were considered statistically significant.

table, older age (OR = 1.03, 95% CI = 1.03–1.04, $p < 0.001$), shortness of breath (OR = 1.88, 95% CI = 1.60–2.20, $p < 0.001$) and chronic kidney disease (OR = 2.41, 95% CI = 1.67–3.48, $p < 0.001$) were estimated to be the risk factors most associated with the mortality of hospitalized COVID-19 patients, while sore throat (OR = 0.74, 95% CI = 0.58–0.95, $p = 0.02$), longer length of hospital stay (OR = 0.96, 95% CI = 0.95–0.97, $p < 0.001$) and time from diagnosis to definitive outcome (OR = 0.97, 95% CI = 0.96–0.98, $p < 0.001$) were more associated with a reduced risk of mortality from COVID-19.

In this work, we are not only interested in evaluating whether or how much the collected variables are correlated to the risk of mortality from COVID-19, but we are also interested in whether those variables, collected for hospitalized patients, are correlated with the length of hospital stay, and the time from diagnosis to the definitive outcome. In Table 3, we show a matrix with p-values estimated using one-way ANOVA and Pearson correlation for those variables.

As shown in Table 3, the relation between age, length of hospital stay and time from diagnosis to definitive outcome is statistically significant ($p < 0.001$). From the Pearson correlation matrix shown in Fig 2, we can see that the length of hospital stay has a moderate positive relationship (Pearson's $r = 0.6$) with the time from diagnosis to definitive outcome, which makes sense, since the longer the patient is hospitalized, the longer is the time to reach an outcome of death or recovery. On the other hand, a weak negative relationship was measured between age and both length of hospital stay and time from diagnosis to definitive outcome (Pearson's $r = -0.1$). Hence, we can see a weak indication that the lower the age of a patient, the longer the patient would stay hospitalized. We could justify that assumption following the results from Table 1, where the mean age of the recovered patients is lower than the mean age of the dead patients (57.7 vs 68.8, $p < 0.001$), but the mean of both the length of hospital stay (16.1 vs 10.1, $p < 0.001$) and time from diagnosis to definitive outcome (24.7 vs 17.2, $p < 0.001$) were longer.

From Table 3, we can also see that, for a few categorical variables (*e.g.*, race, shortness of breath, cough, and low-risk pregnancy), we can provide a valid correlation with respect to the

**Table 3. A matrix of p-values between hospitalized patients' demographics, symptoms, medical history, length of hospital stay and time from diagnosis to definitive outcome.**

| Variables | Length of hospital stay | Time from diagn. to outc. |
|---|---|---|
| Age | < 0.001 | < 0.001 |
| Sex | 0.5 | 0.2 |
| Race | < 0.001 | < 0.001 |
| **Symptoms** | | |
| Sore throat | 0.4 | 0.1 |
| Shortness of breath | < 0.001 | < 0.001 |
| Fever | 0.3 | 0.3 |
| Cough | 0.6 | 0.02 |
| **Comorbidities** | | |
| Diabetes | 0.4 | 0.1 |
| Immunosuppression | 1 | 0.3 |
| Chronic kidney disease | 0.9 | 0.8 |
| Chronic respiratory disease | 0.07 | 0.2 |
| Cardiovascular disease | 0.4 | 0.2 |
| Chromosomal disorder | 0.5 | 0.2 |
| **Others** | | |
| Low-risk pregnancy | **0.001** | **0.02** |
| High-risk pregnancy | 0.3 | 0.3 |
| Length of hospital stay | < 0.001 | < 0.001 |
| Time from diagn. to outc. | < 0.001 | < 0.001 |

Pearson correlation was used to estimate p-values for age, length of hospital stay and time from diagnosis to definitive outcome variables. Age is reported in years. Length of hospital stay and time from diagnosis to definitive outcome are reported in days. The remaining p-values were estimated using one-way ANOVA. Immunosuppression may refer to a patient with HIV infection or autoimmune disease. Bold p-values indicate statistically significant p-values lower than 0.05. "Time from diagn. to outc." is an abbreviation for "Time from diagnosis to definitive outcome".

length of hospital stay or the time from diagnosis to definitive outcome. Since these last two variables were further determined to have a moderate positive relationship, as explained previously, from now on we will focus on the analysis of the relation between those categorical variables and the length of hospital stay, when the variable was determined to be statistically significant for both characteristics of the COVID-19 pandemic.

In Fig 3, we can see that the length of hospital stay is pretty similar regardless of whether a hospitalized patient reported a symptom of shortness of breath ($p < 0.001$). Similarly, from the box plots shown in Fig 4, we can state that an initial symptom of cough does not affect significantly the total time from diagnosis to definitive outcome of COVID-19 for the hospitalized patients ($p = 0.02$).

In Fig 5, we can see that female patients with low-risk pregnancy stayed, on average, a slight longer time at the hospital than the female patients that were not pregnant or did not have a low-risk pregnancy ($p = 0.001$). Despite this fact, we recall from Table 1, that the frequency of low-risk pregnants that recovered from COVID-19 was higher compared to the frequency of low-risk pregnants that died from COVID-19 (1.7% vs 0.4%, $p < 0.001$).

Finally, from Fig 6, we see that race is another variable that does not have a strong relationship with the length of hospital stay, since the mean length of hospital stay is nearly the same regardless of the reported race ($p < 0.001$).

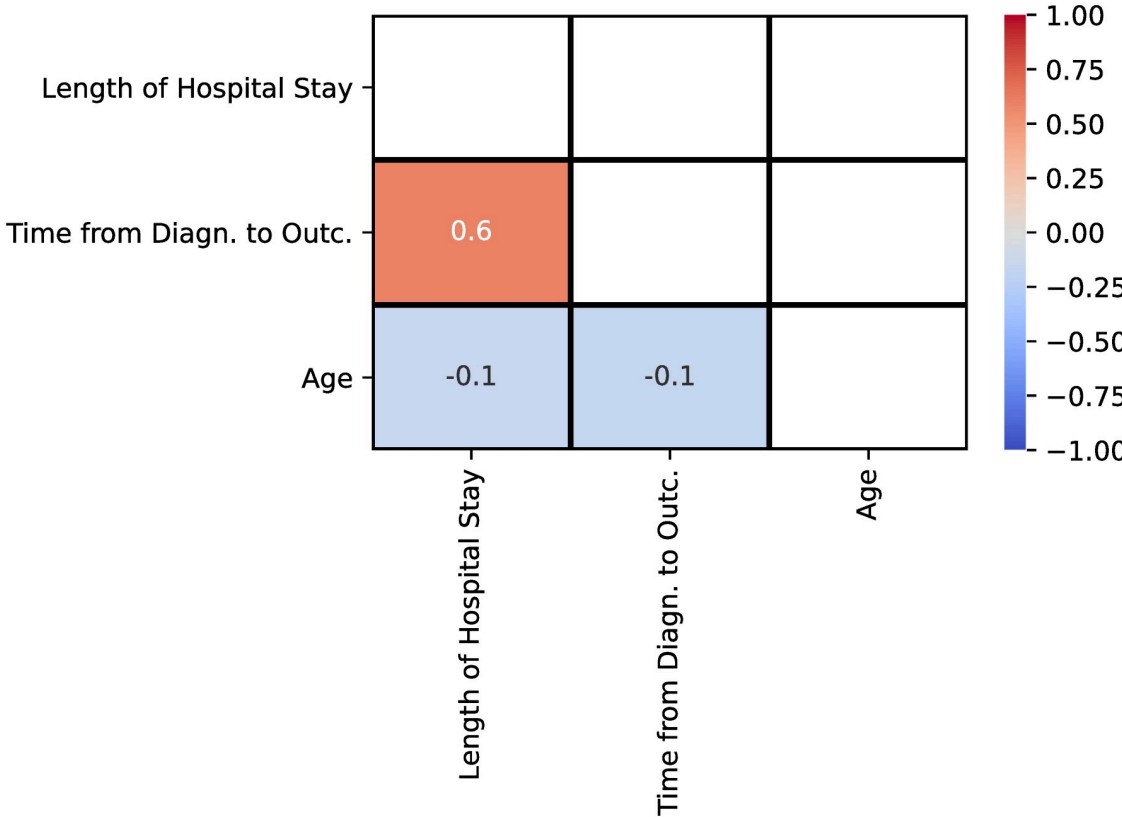

**Fig 2. Pearson correlation matrix for age, length of hospital stay and time from diagnosis to definitive outcome variables.** "Time from diagn. to outc." is an abbreviation for "Time from diagnosis to definitive outcome".

In Table 4, we show the results obtained with a multivariable linear regression analysis performed between the statistically significant variables measured in Table 3 and the length of hospital stay. Similarly to the result reported in Fig 2, older age (OR = 0.97, 95% CI = 0.96–0.99, $p < 0.001$) was also determined to have a weak negative relationship with the length of hospital stay. As shown in Fig 6, patients with an unknown race (OR = 18.03, 95% CI = 4.56–71.73, $p < 0.001$) identification had a higher length of hospital stay than the others, a fact that was captured by the multivariable regression analysis. The initial symptom of shortness of breath ($p = 0.4$) and a low-risk pregnancy ($p = 0.1$) were not estimated to be risk factors for an increased length of hospital stay. Finally, in accordance with the correlation matrix shown in Fig 2, time from diagnosis to definitive outcome (OR = 1.67, 95% CI = 1.63–1.70, $p < 0.001$) has a moderate relation to the length of hospital stay.

## Discussion

In a general sense, our results shown in the previous section point out to the fact that except for the variables age (OR = 0.97, 95% CI = 0.95–0.98, $p < 0.001$, Pearson's $r = -0.1$) and time from diagnosis to definitive outcome (OR = 1.67, 95% CI = 1.64-1.71, $p < 0.001$, Pearson's $r = 0.6$), we cannot guarantee that the other measured variables of the hospitalized patients' demographics, symptoms and comorbidities can differentiate the groups of patients who had a shorter or a longer length of hospital stay.

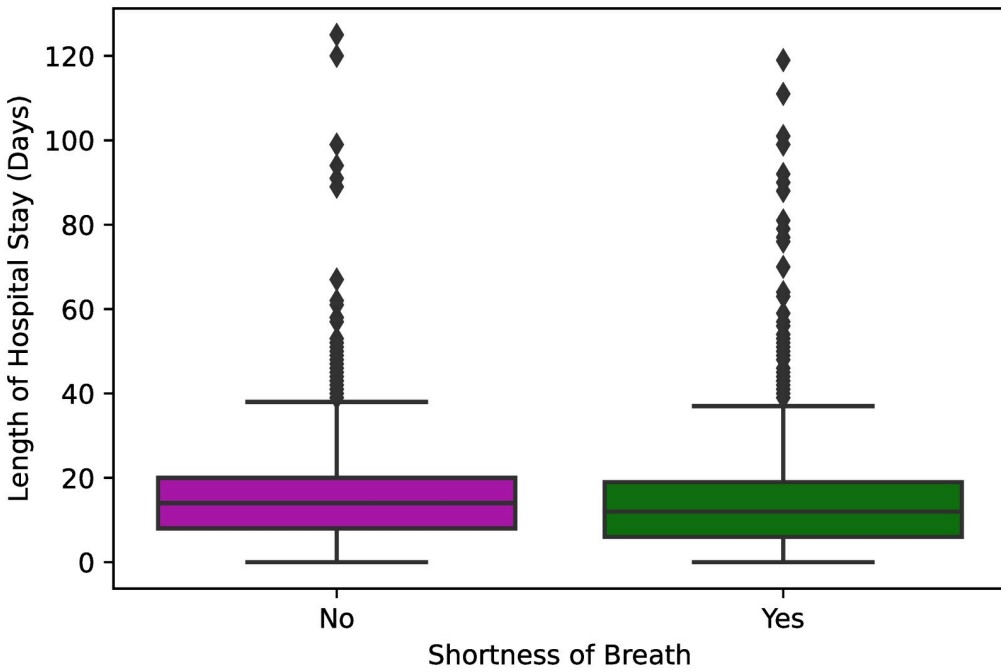

**Fig 3. Box plot showing the length of hospital stay between hospitalized COVID-19 patients that reported and did not report shortness of breath as an initial symptom of COVID-19.** One-way ANOVA was used to estimate a significant p-value <0.001 for those variables.

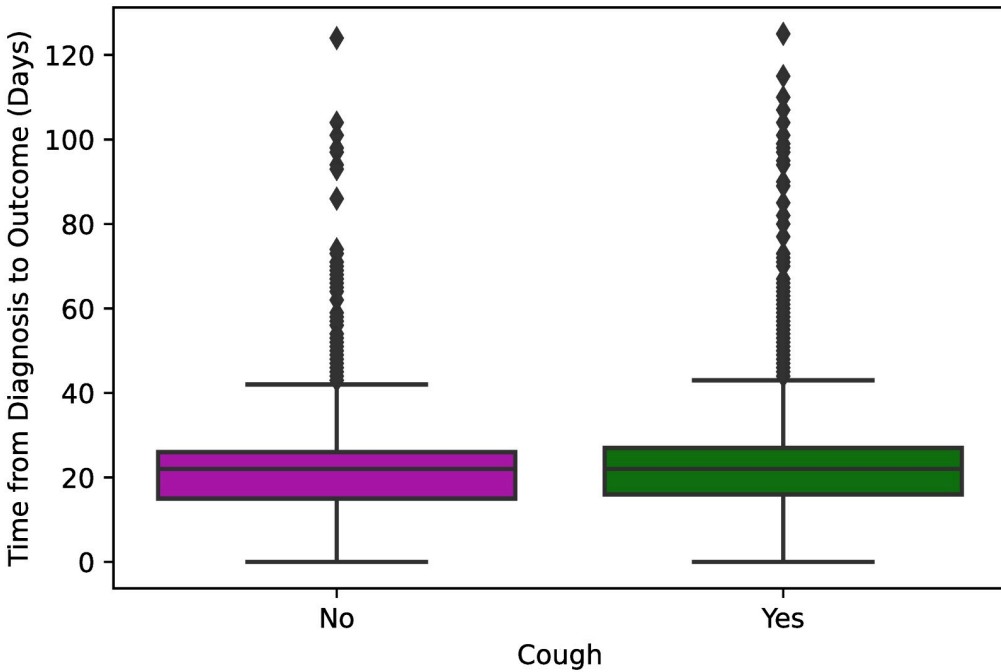

**Fig 4. Box plot showing the time from diagnosis to definitive outcome between hospitalized COVID-19 patients that reported and did not report cough as an initial symptom of COVID-19.** One-way ANOVA was used to estimate a significant p-value <0.001 for those variables.

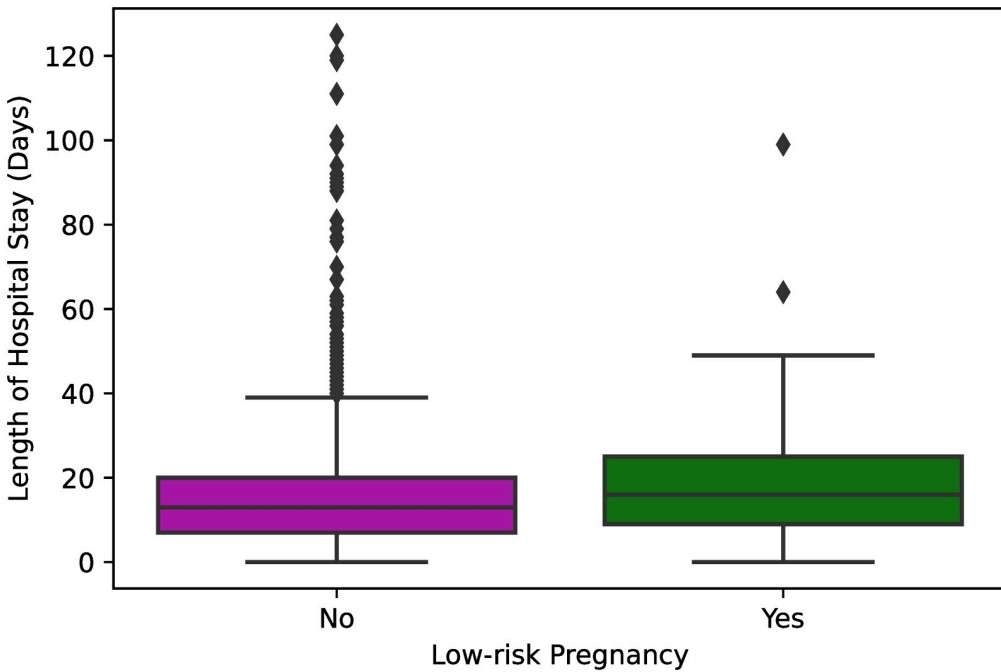

**Fig 5. Box plot showing the length of hospital stay between hospitalized COVID-19 female patients that had and did not have low-risk pregnancy.** One-way ANOVA was used to estimate a significant p-value = 0.001 for those variables.

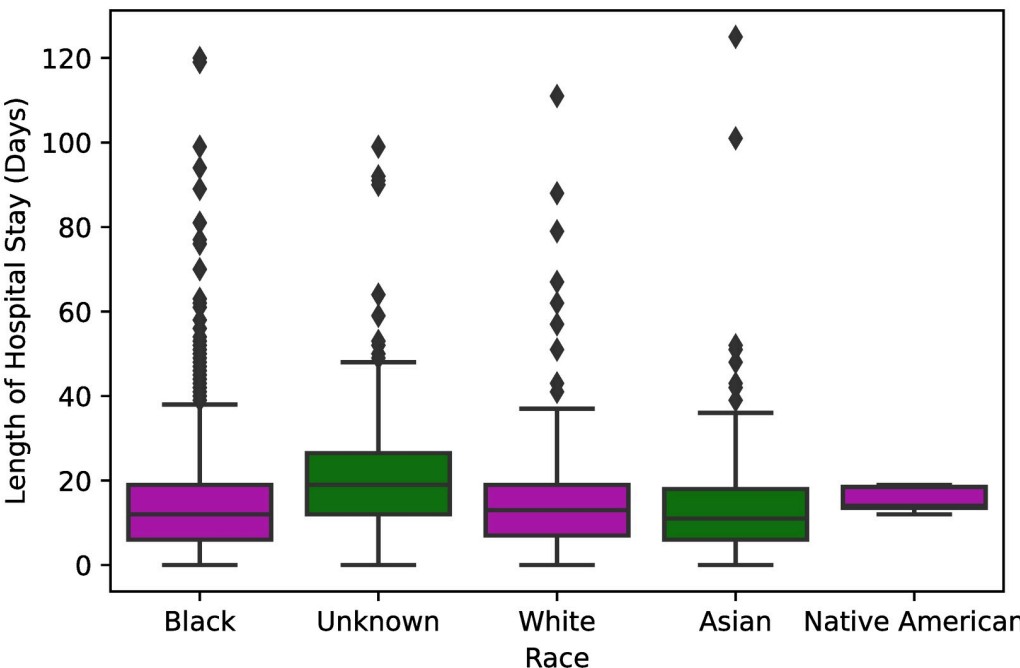

**Fig 6. Box plot showing the length of hospital stay per race of the hospitalized COVID-19 patients.** One-way ANOVA was used to estimate a significant p-value <0.001 for those variables.

**Table 4. Risk factors associated with the length of hospital stay for hospitalized COVID-19 patients.**

| Variables | OR (95% CI) | p-value |
|---|---|---|
| Age | 0.97 (0.96–0.99) | **< 0.001** |
| **Race** | | |
| Black | 0.63 (0.19–2.07) | 0.4 |
| White | 0.73 (0.19–2.78) | 0.6 |
| Asian | 0.47 (0.13–1.76) | 0.3 |
| Native American | 28.36 (0.11–7471.53) | 0.2 |
| Unknown | 18.03 (4.56–71.73) | **< 0.001** |
| **Symptoms** | | |
| Shortness of breath | 1.27 (0.72–2.26) | 0.4 |
| **Others** | | |
| Low-risk pregnancy | 7.82 (0.66–92.97) | 0.1 |
| Time from diagnosis to definitive outcome | 1.67 (1.63–1.70) | **< 0.001** |

Multivariable linear regression was used to measure odds ratios (OR), confidence intervals (CI) and p-values for the statistically significant variables evaluated in Table 3 with respect to the length of hospital stay for hospitalized COVID-19 patients. Bold p-values lower than 0.05 were considered statistically significant.

The box plots for most of the statistically significant categorical variables obtained in the dataset, namely shortness of breath, cough, and race, were nearly the same regardless of the patient's race or whether the patients had the corresponding initial symptom. On the other hand, we have a weak negative relationship between age and length of hospital stay, in the sense that younger people tend to stay more time in the hospital, which may be caused due to the fact that younger patients have a higher chance to recover from COVID-19 than older patients. Moreover, the box plot of Fig 5 shows that low-risk pregnants were found to stay more time hospitalized, although that results were not measured to be statistically significant in the multivariable linear regression (Table 4). Regarding the pregnancy criterion, there is current evidence of low rates of maternal and neonatal mortality, as well as of Intensive Care Unit admissions associated with COVID-19 [20].

It is noteworthy that the fever symptom appears as an important indicator in the screening of suspected or confirmed cases of COVID-19, despite not having, in this study, a strong relationship with the length of hospital stay. Liu et al. [21], in a research to determine the risk factors for severe COVID-19, report that fever, cough and fatigue appear as the main symptoms among the participants, with fever being the most common initial symptom (92.3%). Rivera-Izquierdo et al. [22] complement that in patients admitted for COVID-19 at Hospital Universitario Clínico San Cecilio, Spain, the most frequent clinical picture was also due to low fever (89.5%) and dry cough (80.7%), followed by general malaise (63.5%), dyspnea (61.3%) and tiredness (59.2%).

With respect to the statistically significant factors that could be correlated with the mortality of COVID-19 patients, Table 1 shows that non-survivor patients had a higher mean age than survivor patients. Moreover, the frequency of non-surviving patients that reported shortness of breath as an initial symptom of COVID-19 was higher than the frequency reported by survivor patients. Finally, most of the comorbidities were more frequent in non-survivor patients. A study carried out by Zhang et al. [23] with patients admitted to Renmin Hospital, Wuhan University, and with a laboratory-confirmed diagnosis of COVID-19, recorded that the majority of patients who died were male, over 60 years old, and had comorbidities.

Comparing our results evaluated in the state of Bahia, Brazil, with the reported results of other regions of the world, we see that, in an other state of Brazil, namely Espírito Santo, Soares *et al.* [19] reported a higher mortality rate of 39.6% of the hospitalized patients, when compared to the mortality rate reported in this study (26.8%). Moreover, similar to our results, older age, shortness of breath and the kidney disease were also estimated to be risk factors associated with the mortality from COVID-19. A more general study conducted by Souza *et al.* [24] accounting for 208,969 hospitalized patients in Brazil, shows that the mortality rate in this country is high considering only the hospitalized patients (41.28%), and, older age, shortness of breath, the presence of comorbidities, and the ventilation requirement were more associated with the mortality of those patients. Looking at the results obtained in another country, the work of Imam *et al.* [12] reported that only 15.3% of the hospitalized patients in Michigan, United States, did not survive from COVID-19. While that mortality rate is lower than the ones estimated in Brazil, older age and comorbidities were again measured to be risk factors associated with the mortality from COVID-19. Similar results were reported in a more general analysis performed with data collected from 154 hospitals distributed in 13 states of the United States [25]. Using data collected in Wuhan, China, Cheng *et al.* [26] evaluated that 16.1% of the hospitalized patients died from COVID-19, and the kidney disease was determined to be one of main risk factor associated with the mortality of those patients.

Finally, we have analyzed whether pairs of combined variables (*e.g.*, fever + cough, diabetes + cardiovascular disease) could influence in the risk of mortality, length of hospital stay and time from diagnosis to definitive outcome of COVID-19. However, the obtained p-values were higher than the default threshold.

As threats to the validity of this work, we can state that all the results available in the dataset provided by SESAB were reported by the own patients, their families or the healthcare professionals working in the hospitals. Nevertheless, we have used the official dataset provided by the government of the state of Bahia, in Brazil, which further reinforces the relevance of this work. Moreover, as shown in Fig 1, we performed a careful data validation to remove outliers or patients with invalid data, restricting our analysis for valid hospitalized patients. Again, to favor openness and reproducibility, we provide the anonymized dataset as supporting information of this manuscript.

## Conclusion

COVID-19 is affecting millions of people around the world, with a mortality rate of about 2–3% of all the infected patients. Due to the ease of spread of this disease, health systems are challenged to guarantee universal access to their healthcare services, while handling the high demands that may overload their capacity limit. In this sense, we have provided, in this work, a retrospective cohort study with 3,896 patients of COVID-19 that have been hospitalized in the state of Bahia, Brazil, in order to assist our health care systems with respect to an analysis of which variables could influence in the mortality rate, length of hospital stay, and time from diagnosis to definitive outcome of COVID-19. This data exploratory analysis is fundamental to aid these organizations in the decision-making process, enabling them to improve their services to the infected population. Therefore, we could estimate that older age, presence of shortness of breath, and chronic kidney disease were the variables associated with an increased risk of mortality from COVID-19. On the other hand, the presence of sore throat and a higher length of hospital stay seems to not increase the risk of mortality from COVID-19. Moreover, only younger age and a higher time from diagnosis to definitive outcome were the variables associated with a higher length of hospital stay.

For future work, since the COVID-19 is a recent disease and the pandemic is still active, with new patients being hospitalized every day, one could estimate the relation between the evaluated variables for a larger number of hospitalized patients, to see whether new correlations arise in an increased cohort population. Moreover, we aim to further assist the health systems in their decision-making process by developing a new technological solution to predict length of hospital stay given the patients' demographics, symptoms and comorbidities.

## Supporting information

**S1 Table. Anonymized dataset of hospitalized patients from COVID-19 in the state of Bahia.**
(CSV)

## Acknowledgments

The authors would like to thank SESAB for providing the anonymized dataset of COVID-19 patients in the state of Bahia, Brazil.

## Author Contributions

**Conceptualization:** Isabelle M. Pinheiro, Hilda C. J. R. Fraga, Simone S. Montes, Ivonete T. L. Queiroz, Romilson L. Sampaio, Antonio C. S. Souza.

**Data curation:** Caio J. L. Carvalho, Isaac P. C. Araujo, Otávio A. C. Araujo, Lucas A. Alves, Márcio L. V. Araújo, Márcia S. P. L. Souza, Ana Claudia F. N. da Silva.

**Formal analysis:** Márcio C. F. Macedo, Caio J. L. Carvalho.

**Funding acquisition:** Antonio C. S. Souza.

**Investigation:** Caio J. L. Carvalho, Isaac P. C. Araujo, Otávio A. C. Araujo, Lucas A. Alves.

**Methodology:** Márcio C. F. Macedo, Caio J. L. Carvalho, Hugo Saba, Márcio L. V. Araújo.

**Project administration:** Márcio C. F. Macedo, Antonio C. S. Souza.

**Resources:** Caio J. L. Carvalho.

**Software:** Márcio C. F. Macedo, Caio J. L. Carvalho, Isaac P. C. Araujo, Otávio A. C. Araujo, Lucas A. Alves.

**Supervision:** Márcio C. F. Macedo, Hugo Saba, Márcio L. V. Araújo, Antonio C. S. Souza.

**Validation:** Márcio C. F. Macedo, Isabelle M. Pinheiro, Caio J. L. Carvalho, Hilda C. J. R. Fraga, Simone S. Montes, Márcio L. V. Araújo, Ivonete T. L. Queiroz, Márcia S. P. L. Souza, Ana Claudia F. N. da Silva.

**Visualization:** Márcio C. F. Macedo, Caio J. L. Carvalho.

**Writing – original draft:** Márcio C. F. Macedo, Isabelle M. Pinheiro, Hilda C. J. R. Fraga, Simone S. Montes, Ivonete T. L. Queiroz.

**Writing – review & editing:** Márcio C. F. Macedo, Isabelle M. Pinheiro, Hugo Saba, Márcio L. V. Araújo, Romilson L. Sampaio, Márcia S. P. L. Souza, Ana Claudia F. N. da Silva, Antonio C. S. Souza.

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
