## [Decision Letter · Decision Letter 0]

21 Oct 2020

PONE-D-20-29331

Correlation between hospitalized patients' demographics, symptoms, comorbidities, and COVID-19 pandemic in Bahia,

Brazil

PLOS ONE

Dear Dr. Macedo,

Thank you for submitting your manuscript to PLOS ONE. After careful consideration, we feel that it has merit but does not fully meet PLOS ONE’s publication criteria as it currently stands. Therefore, we invite you to submit a revised version of the manuscript that addresses the points raised during the review process.

ACADEMIC EDITOR: Please review the comments made by the reviewers and provide a point by point response in your revised manuscript. 

We look forward to receiving your revised manuscript.

Kind regards,

Muhammad Adrish

Academic Editor

PLOS ONE

Journal Requirements:

Reviewers' comments:

Reviewer's Responses to Questions

**Comments to the Author**

1. Is the manuscript technically sound, and do the data support the conclusions?

Reviewer #1: Yes

Reviewer #2: Partly

2. Has the statistical analysis been performed appropriately and rigorously? 

Reviewer #1: Yes

Reviewer #2: No

3. Have the authors made all data underlying the findings in their manuscript fully available?

Reviewer #1: No

Reviewer #2: No

4. Is the manuscript presented in an intelligible fashion and written in standard English?

Reviewer #1: Yes

Reviewer #2: Yes

5. Review Comments to the Author

Reviewer #1: This is a straightforward presentation of clinical data from a COVID-19 patient cohort in Brazil and provides some associations between the outcome of death and demographic parameters/selected key symptoms. A limitation is that no laboratory clinical parameters are provided so the study could be improved if there are any laboratory findings that could also be included, even for a subset of patients. In general, I think this is an important set of data from a region of the world where there are few publications relating to COVID-19 so it would be informative when included in the literature after some of the major data presentation issues are rectified.

Key points for revision:

The percentage of deaths should be clearly given in the text early on in the results section. Is it possible that better records were provided for patients who were more severe and/or died and many of the incomplete or incorrect records were for milder patients? Because it seems to be a very high death rate.

table 1 is useful because it provides the outcomes, but it is unclear why the percentages don’t add up. Are there patients who were still being monitored at the time of submission? In other words, are there unresolved cases that are not indicated here?

Also in table 1, what are the numbers in parentheses beside the average age? From the table above it suggests that it should be the n, but that is not possible.

The definition of “race” is not standard and seems politically incorrect in English. For example words such as “yellow” have negative connotations and can be interpreted as racist. I think more standard terms need to be given rather than the colors which might be a direct translation but do not sound appropriate in English.

Perhaps in the results the kinds of immunosuppression could be given? Is this due to chronic drug use by those patients or does it also include HIV infections?

It would be helpful since the numbers of low and high risk pregnancy are few, to add a category of total pregnancy to the list? Were pregnant women more likely to be hospitalized, potentially explaining their better outcomes?

For Table 2, significant p-values could be highlighted or otherwise indicated to aid readability.

For Fig 1. Please use commas rather than decimals in the numbers (e.g. 3.896 -> 3,896)

For Figs 3-6 (which were out of order in the file, please correct next time) It would be good to put an indication on the figure of the p-value and/or if it was significant. The figure legends need to contain the statistical test used for all figures.

The discussion should comment on the high mortality rate of the selected group of patients and how this compares to the overall mortality rate in Brazil. Also, some comparison to studies that were done in other regions where similar or different observations were made would greatly improve the discussion.

Reviewer #2: There are a number of concerns regarding how the analysis was presented/conducted. Given that hospital length of stay and definitive outcome are going to be high-correlated for patients hospitalized for COVID-19, it would seem best to treat length of stay, death, and recovery as competing outcomes and to analyze the data accordingly. I would also like to see more efforts to account for the potential confounding effects that different characteristics might have on each other, possibly using a multivariable regression approach to that analysis.

6. PLOS authors have the option to publish the peer review history of their article (what does this mean?). If published, this will include your full peer review and any attached files.

Reviewer #1: No

Reviewer #2: No

---

## [Author Response · Author response to Decision Letter 0]

29 Oct 2020

Dear associate editor and reviewers, thank you very much for your assistance and reviews. In the letter of rebuttal submitted together with this revised version of the manuscript, we provide the responses to each point that you raised. Moreover, we uploaded the anonymized dataset that we have used in this work as a supporting information file of this submission. Thank you very much!

---

## [Decision Letter · Decision Letter 1]

25 Nov 2020

PONE-D-20-29331R1

Correlation between hospitalized patients' demographics, symptoms, comorbidities, and COVID-19 pandemic in Bahia, Brazil

PLOS ONE

Dear Dr. Macedo,

Thank you for submitting your manuscript to PLOS ONE. After careful consideration, we feel that it has merit but does not fully meet PLOS ONE’s publication criteria as it currently stands. Therefore, we invite you to submit a revised version of the manuscript that addresses the points raised during the review process.

ACADEMIC EDITOR: Please see attached comments by the reviewers. Kindly provide point by point response in your revised manuscript.

We look forward to receiving your revised manuscript.

Kind regards,

Muhammad Adrish

Academic Editor

PLOS ONE

Reviewers' comments:

Reviewer's Responses to Questions

**Comments to the Author**

1. If the authors have adequately addressed your comments raised in a previous round of review and you feel that this manuscript is now acceptable for publication, you may indicate that here to bypass the “Comments to the Author” section, enter your conflict of interest statement in the “Confidential to Editor” section, and submit your "Accept" recommendation.

Reviewer #1: All comments have been addressed

Reviewer #2: All comments have been addressed

2. Is the manuscript technically sound, and do the data support the conclusions?

Reviewer #1: Yes

Reviewer #2: (No Response)

3. Has the statistical analysis been performed appropriately and rigorously? 

Reviewer #1: Yes

Reviewer #2: (No Response)

4. Have the authors made all data underlying the findings in their manuscript fully available?

Reviewer #1: Yes

Reviewer #2: (No Response)

5. Is the manuscript presented in an intelligible fashion and written in standard English?

Reviewer #1: Yes

Reviewer #2: (No Response)

6. Review Comments to the Author

Reviewer #1: The authors have addressed my comments and I only had one minor confusion remaining. Which race category are the authors using for native american populations? I wasn't sure if they used "Indian" for this and if so, it might be better to change to "native american" to avoid confusion with India as a country of origin.

Reviewer #2: (No Response)

7. PLOS authors have the option to publish the peer review history of their article (what does this mean?). If published, this will include your full peer review and any attached files.

Reviewer #1: No

Reviewer #2: No

---

## [Author Response · Author response to Decision Letter 1]

25 Nov 2020

We thank the reviewer #1 for his/her suggestion and, indeed, we have used the term “Indian” to describe “Native American”. In this revised version of the manuscript, we have followed the suggestion of the reviewer and we changed “Indian” to “Native American” to avoid confusion.

---

## [Decision Letter · Decision Letter 2]

2 Dec 2020

Correlation between hospitalized patients' demographics, symptoms, comorbidities, and COVID-19 pandemic in Bahia, Brazil

PONE-D-20-29331R2

Dear Dr. Macedo,

We’re pleased to inform you that your manuscript has been judged scientifically suitable for publication and will be formally accepted for publication once it meets all outstanding technical requirements.

Kind regards,

Muhammad Adrish

Academic Editor

PLOS ONE

Additional Editor Comments (optional): You have satisfactorily answered all the queries raised by the reviewers. 

Reviewers' comments:

Reviewer's Responses to Questions

**Comments to the Author**

1. If the authors have adequately addressed your comments raised in a previous round of review and you feel that this manuscript is now acceptable for publication, you may indicate that here to bypass the “Comments to the Author” section, enter your conflict of interest statement in the “Confidential to Editor” section, and submit your "Accept" recommendation.

Reviewer #1: All comments have been addressed

2. Is the manuscript technically sound, and do the data support the conclusions?

Reviewer #1: Yes

3. Has the statistical analysis been performed appropriately and rigorously? 

Reviewer #1: Yes

4. Have the authors made all data underlying the findings in their manuscript fully available?

Reviewer #1: Yes

5. Is the manuscript presented in an intelligible fashion and written in standard English?

Reviewer #1: Yes

6. Review Comments to the Author

Reviewer #1: All of my comments were addressed. I have no further comments. I recommend accepting the manuscript.

7. PLOS authors have the option to publish the peer review history of their article (what does this mean?). If published, this will include your full peer review and any attached files.

Reviewer #1: No

---

## [Editor Report · Acceptance letter]

4 Dec 2020

PONE-D-20-29331R2 

Correlation between hospitalized patients’ demographics, symptoms, comorbidities, and COVID-19 pandemic in Bahia, Brazil  

Dear Dr. Macedo:

I'm pleased to inform you that your manuscript has been deemed suitable for publication in PLOS ONE. Congratulations! Your manuscript is now with our production department. 

Kind regards, 

on behalf of

Dr. Muhammad Adrish 

Academic Editor

PLOS ONE